# Design, Synthesis and Evaluation of Novel 1,4-Disubstituted Piperazine-2,5-dione Derivatives as Antioxidants against H_2_O_2_-Induced Oxidative Injury via the IL-6/Nrf2 Loop Pathway

**DOI:** 10.3390/antiox11102014

**Published:** 2022-10-12

**Authors:** Liang Xiong, Hongshan Wu, Ting Zhong, Fang Luo, Qing Li, Mei Li, Yanhua Fan

**Affiliations:** 1State Key Laboratory for Functions and Applications of Medicinal Plants, Guizhou Medical University, Guiyang 550014, China; 2The Key Laboratory of Chemistry for Natural Products of Guizhou Province and Chinese Academy of Sciences, Guiyang 550014, China

**Keywords:** 1,4-disubstituted piperazine-2,5-diones, oxidative stress, cell apoptosis, cell survival, IL-6/Nrf2 positive-feedback loop

## Abstract

Excessive reactive oxygen species (ROS) production leads to oxidative stress in cells, impairing the function of mitochondria and finally inducing cell apoptosis. Considering the essential role of oxidative stress in the pathogenesis of various neurodegenerative diseases and psychiatric disorders, the discovery of novel antioxidants has attracted increasing attention. Herein, a series of novel 1,4-disubstituted piperazine-2,5-dione derivatives were designed, synthesized and evaluated for their antioxidative activity. The results of the 3-(4,5-dimethylthiazol-2-yl)-2,5-diphenyl-2H-tetrazolium bromide (MTT) assay indicated that none of the tested compounds showed significant toxicity to SH-SY5Y cells at concentrations up to 80 μM. Cell counting via flow cytometry revealed that most of the tested compounds could effectively protect SH-SY5Y cells from H_2_O_2_-induced oxidative damage at 20 μM. Among these compounds, compound **9r** exhibited the best antioxidative activity. Further mechanistic investigation indicated that **9r** decreased ROS production and stabilized the mitochondrial membrane potential to restrain cell apoptosis, and promoted cell survival via an IL-6/Nrf2 positive-feedback loop. These results suggested the potential of compound **9r** as a novel antioxidative candidate for the treatment of diseases caused by oxidative stress.

## 1. Introduction

Oxidative stress, caused by the unbalanced production of reactive oxygen species (ROS) and antioxidants, is a contributing factor in the onset and progression of many diseases, including neurodegenerative disorders, stroke and inflammation [1]. Under physiological conditions, the level of ROS formation is in equilibrium with the antioxidant capacity. However, supraphysiological levels of ROS lead to oxidative stress and subsequent damage to biomolecules [2]. ROS are produced and result in oxidative stress when cells are subjected to the stimulation of superoxide, hydrogen peroxide and hydroxyl free radicals [3]. Among these species, hydrogen peroxide (H_2_O_2_) is responsible for redox signaling transmission from the generative site to a target site. Hence, H_2_O_2_ is considered most suitable for the establishment of the oxidative stress model.

Usually, cells resist oxidative stress via the action of antioxidants as well as the activation of antioxidative enzymes, including catalase, glutathione peroxidase and glutathione peroxidase. The transcription factor nuclear factor erythroid-2 related factor 2 (Nrf2) is one of the crucial antioxidant pathways. Under normal conditions, most Nrf2 is kept in an inactive state in the form of complexes with Kelch-like ECH-associated protein 1 (Keap1) in the cytoplasm and is constantly targeted for degradation via ubiquitination. However, when cells are undergoing oxidative stress, the complex between Nrf2 and Keap1 depolymerizes and leads to the accumulation of Nrf2 in the cytoplasm. The ARE (the antioxidant response element) is activated when Nrf2 translocates to the nucleus and binds to the small muscular aponeurotic fibrosarcoma (Maf) protein. The activated ARE further leads to the transcriptional activation of various antioxidant enzymes and proteins, including haem oxygenase-1 (HO-1) and prosurvival cytokines (e.g., interleukin (IL)-6) [4]. IL-6, a pleiotropic cytokine, plays an important role in cell survival, proliferation, immunity and inflammation. When cells are under oxidative stress, Nrf2 mediates IL-6 expression by indirectly activating its promoter through the ARE and exerts partial antioxidant activity. In turn, IL-6 reduces cell apoptosis and promotes cell survival by activating downstream signaling, including MAPKs, JAK/Stat3 and PI3K/Akt, and increasing the nuclear translocation or stability of Nrf2 [5,6,7,8]. Hence, the development of antioxidants targeting the IL-6/Nrf2 cross axis may provide a novel therapeutic strategy for the treatment of oxidative stress-related diseases.

Natural bisepoxylignans exhibit diverse biological activities, including antioxidant [9] and anti-inflammatory [10] activities. Furofuran lignans are naturally occurring bisepoxylignans abundant in the plant kingdom with diverse biological activities, including antioxidative properties [11]. Previous studies have indicated that 137 furofuran lignans have been isolated and identified from various plants. Among these compounds, sesamin, syringaresinol, terminaloside K and pinoresinol showed significant antioxidant activities [12]. In our previous study, we also isolated and identified several known bisepoxylignans, including syringaresinol, from the roots of *Zea mays* L. (maize) and found that they protected SH-SY5Y cells against H_2_O_2_-induced cell apoptosis [13]. Therefore, bisepoxylignans are considered a good source of natural antioxidants [14]. However, due to their multiple chiral centers (usually more than 4) [15,16], it is difficult to synthesize these antioxidant active compounds. To date, there are few active compounds that can be synthesized in large quantities. Hence, it is still necessary to perform structural modification to retain their antioxidative activity and make them easy to synthesize. 2,5-piperazinedione, prepared via the condensation and cyclization of two molecules of glycine, is the smallest cyclic peptide derivative in nature. It is attracting increasing attention due to its unique chemical properties, including strong rigidity and exceptionally high stability to peptidase, and its biological characteristics, including an extensive range of biological activities. Scaffold hopping, an important and efficient drug-design strategy for molecular-backbone replacements, has been widely used in drug design to develop novel molecules with potent activity [17]. Heterocycle replacements are one of the commonly used methodologies that enable the generation of scaffold hops [18]. Herein, we designed and synthesized a series of novel 1,4-disubstituted piperazine-2,5-diones as derivatives of bisepoxylignans based on a scaffold-hopping strategy (Figure 1) and evaluated their antioxidative activities using a H_2_O_2_-induced oxidative stress model. The potential antioxidant mechanisms of the best active compound, **9r**, were further investigated.

## 2. Materials and Methods

### 2.1. General Methods 

All reagents and solvents used in the experiment were commercially available and used without further purification. ^1^H nuclear magnetic resonance (NMR) data were measured and recorded on a Bruker 600 MHz NMR spectrometer (Bruker; Billerica, MA, USA), and ^13^C NMR spectral data were measured and recorded using a Bruker 151 MHz NMR spectrometer with chloroform-d as the solvent and tetramethylsilane (TMS) as the internal standard. The high-resolution mass spectra (HRMS) of all target compounds were measured using a mass spectrometer (microTOF-Q, Bruker Inc.) using electrospray ionization. The melting points were measured using a melting-point instrument (Shanghai Micro Melting Point Apparatus, Shanghai, China), and the values are uncorrected.

### 2.2. General Method for the Synthesis of Compounds **9a**–**9s**

#### 2.2.1. Ethyl 2-[(3,5-Dimethoxyphenyl)amino]acetate (**2a**)

Under a N_2_ atmosphere, a mixture of 1a (100 mg, 0.653 mmol) and anhydrous K_2_CO_3_ (135 mg, 0.977 mmol) in anhydrous acetone (5 mL) was heated to 60 °C and stirred for 1 h. A solution of ethyl bromoacetate (142 mg, 0.850 mmol) in anhydrous acetone (1 mL) was added dropwise, and the mixture was stirred at 60 °C for 8 h (thin-layer chromatography (TLC) monitoring). The K_2_CO_3_ was filtered off, and the solvent was removed under vacuum. Then, it was purified via column chromatography over silica gel (PE:EA = 30:1) to give ethyl 2-[(3,5-dimethoxyphenyl)amino]acetate(**2a**) as a yellow oily liquid (134 mg, 0.561 mmol, 86% yield). Electrospray ionization–mass spectrometry (ESI-MS): m/z, 240.1 [M + H]^+^.

#### 2.2.2. Ethyl N-(2-Chloroacetyl)-N-[(3,5-dimethoxyphenyl)amino]acetate (**3a**)

A solution of chloroacetyl chloride (106 mg, 0.939 mmol) was added to ethyl 2-[(3,5-dimethoxyphenyl)amino]acetate (**2a**) (150 mg, 0.627 mmol) and a catalytic amount of tetrabutyl ammonium hydrogen sulfate in CH_2_Cl_2_ (6 mL). A solution of K_2_CO_3_ (130 mg, 0.941 mmol) in H_2_O (2 mL) was slowly added dropwise. The mixture was stirred (TLC monitoring of the reaction). The solvent was removed under vacuum, and the crude product was purified via silica gel column chromatography (PE:EA = 4:1) to give ethyl N-(2-chloroacetyl)-N-[(3,5-dimethoxyphenyl)amino] acetate (**3a**) as a yellow oily liquid (194 mg, 0.614 mmol, 98% yield). ESI-MS: m/z, 316.2, 318.2 [M + H]^+^.

#### 2.2.3. Ethyl N-(2-Azidoacetyl)-N-[(3,5-dimethoxyphenyl)amino]acetate (**4a**)

NaN_3_ (26 mg, 0.400 mmol) was added to a solution of ethyl N-(2-chloroacetyl)-N-[(3,5-dimethoxyphenyl)amino] acetate (**3a**) (50 mg, 0.158 mmol) in DMF (3 mL). The mixture was heated at 90 °C for 1 h (completion monitored via TLC). H_2_O_2_ (5 mL) was added to the mixture. Dimethylformamide (DMF) and H_2_O_2_ were evaporated at 90 °C. The crude product was subjected to silica gel column chromatography (PE:EA = 2:1) to give N-(2-azidoacetyl)-N-[(3,5-dimethoxyphenyl)amino]ethyl ester acetate (**4a**) as a colorless oily liquid (50 mg, 0.155 mmol, 98% yield). ESI-MS: m/z, 323.3 [M + H]^+^.

#### 2.2.4. 1-(3,5-Dimethoxyphenyl)piperazine-2,5-dione (**7a**)

Triphenylphosphine (61 mg, 0.232 mmol) was added to a solution of ethyl N-(2-azidoacetyl)-N-[(3,5-dimethoxyphenyl)amino] acetate (**4a**) (50 mg, 0.155 mmol) in THF (5 mL) with a few drops of water. The mixture was stirred at room temperature for 4 h (monitoring via TLC for completion). The THF was completely evaporated, and the product was purified via silica gel column chromatography using ethyl acetate (EA) as the eluent to give 1-(3,5-dimethoxyphenyl)piperazine-2,5-dione (**7a**) as a white solid (23 mg, 0.092 mmol, 59% yield). ESI-MS: m/z, 251.2 [M + H]^+^.

Compounds **7b**–**7e** were synthesized according to the procedure described in **7a**.

1-(3,4-Methylenedioxyphenyl)piperazine-2,5-dione (**7b**): White Solid, 62% Yield: ESI-MS m/z: 235.2 [M + H]^+^.

1-Phenylpiperazine-2,5-dione (**7c**) [19]: White Solid, 58% Yield: ESI-MS m/z: 191.2 [M + H]^+^.

1-(3-Methoxyphenyl)piperazine-2,5-dione (**7d**): White Solid, 63% Yield: ESI-MS m/z: 221.1 [M + H]^+^.

1-(4-Methoxyphenyl)piperazine-2,5-dione (**7e**) [19]: White Solid, 61% Yield: ESI-MS m/z: 221.1 [M + H]^+^.

#### 2.2.5. 1-(3,5-Dimethoxyphenyl)-4-(4-methoxybenzoyl)piperazine-2,5-dione (**9a**)

4-Methoxybenzoyl chloride (53 mg, 0.312 mmol) in CH_2_Cl_2_ (1 mL) was added dropwise to 1-(3,5-dimethoxyphenyl)piperazine-2,5-dione (**7a**) (60 mg, 0.240 mmol), triethylamine (48 mg, 0.474 mmol) and 4-dimethylaminopyridine (3 mg, 0.024 mmol) in CH_2_Cl_2_ (5 mL) at 0 °C using a dropping funnel. The mixture was stirred for 2 h at room temperature. The completion of the reaction was monitored via TLC. The mixture was poured into 20 mL of saturated brine and extracted with dichloromethane (20 mL × 3). The organic phases were combined and dried with MgSO_4,_ and the solvent was removed under vacuum. The product was obtained via column chromatography over silica gel using PE-EA (2:1) as the eluent to give 1-(3,5-dimethoxyphenyl)-4-(4-methoxybenzoyl)piperazine-2,5-dione (**9a**) as a white solid (74 mg, 0.193 mmol, 80% yield). mp: 180–183 °C. ^1^H NMR (600 MHz, chloroform-d) δ: 7.70 (d, J = 8.8 Hz, 2H), 6.92 (d, J = 8.8 Hz, 2H), 6.48 (d, J = 2.2 Hz, 2H), 6.42 (t, J = 2.2 Hz, 1H), 4.54 (s, 2H), 4.45 (s, 2H), 3.86 (s, 3H), 3.80 (s, 6H). ^13^C NMR (151 MHz, CDCl_3_) δ: 170.47, 166.97, 164.84, 163.81, 161.36, 141.02, 131.85, 125.91, 113.81, 103.51, 99.65, 55.68, 55.62, 54.17, 48.77. HRMS (ESI): calcd for C_20_H_21_O_6_N_2_ [M + H]^+^ m/z, 385.13941; found, 385.13763; C_20_H_20_O_6_N_2_Na [M+Na]^+^ m/z, 407.12136; found 407.12033.

Compounds **9b**–**9s** Were Synthesized According to the Procedure Described in **9a**

#### 2.2.6. 1-(3,5-Dimethoxyphenyl)-4-(4-methylbenzoyl)piperazine-2,5-dione (**9b**)

White solid, 82% yield, mp: 131–134 °C. ^1^H NMR (600 MHz, chloroform-d) δ: 7.59 (d, J = 8.0 Hz, 2H), 7.24 (d, J = 8.0 Hz, 2H), 6.49 (d, J = 2.2 Hz, 2H), 6.43 (t, J = 2.2 Hz, 1H), 4.57 (s, 2H), 4.46 (s, 2H), 3.81 (s, 6H), 2.41 (s, 3H). ^13^C NMR (151 MHz, CDCl_3_) δ: 171.17, 166.88, 164.74, 161.39, 144.09, 141.00, 131.19, 129.26, 129.21, 103.53, 99.69, 55.69, 54.17, 48.58, 21.87. HRMS (ESI): calcd for C_20_H_21_O_5_N_2_ [M + H]^+^ m/z, 369.14450; found, 369.14343; C_20_H_20_O_5_N_2_Na [M + Na]^+^ m/z, 391.12644; found, 391.12540.

#### 2.2.7. 1-(4-Methylbenzoyl)-4-(3,4-methylenedioxyphenyl)piperazine-2,5-dione (**9c**)

White solid, 88% yield, mp: 174–177 °C. ^1^H NMR (600 MHz, chloroform-d) δ: 7.58 (d, J = 8.0 Hz, 2H), 7.24 (d, J = 8.0 Hz, 2H), 6.88–6.83 (m, 2H), 6.74 (dd, J = 8.3, 2.2 Hz, 1H), 6.01 (s, 2H), 4.55 (s, 2H), 4.42 (s, 2H), 2.41 (s, 3H). ^13^C NMR (151 MHz, CDCl_3_) δ: 171.20, 166.86, 164.93, 148.42, 147.19, 144.07, 133.21, 131.20, 129.23, 129.21, 118.50, 108.65, 107.01, 101.99, 54.59, 48.44, 21.87. HRMS (ESI): calcd for C_19_H_17_O_5_N_2_ [M + H]^+^ m/z, 353.11320; found, 353.11182; C_19_H_16_O_5_N_2_Na [M + Na]^+^ m/z, 375.09514; found, 375.09427.

#### 2.2.8. 1-(4-Methylbenzoyl)-4-phenylpiperazine-2,5-dione (**9d**)

White solid, 89% yield, mp: 170–173 °C. ^1^H NMR (600 MHz, chloroform-d) δ: 7.60 (d, J = 8.2 Hz, 2H), 7.46 (t, J = 8.0 Hz, 2H), 7.37–7.31 (m, 3H), 7.25 (d, J = 8.0 Hz, 2H), 4.58 (s, 2H), 4.49 (s, 2H), 2.41 (s, 3H). ^13^C NMR (151 MHz, CDCl_3_) δ: 171.17, 166.94, 164.76, 144.07, 139.37, 131.18, 129.56, 129.25, 129.20, 127.63, 124.83, 54.06, 48.53, 21.86. HRMS (ESI): calcd for C_18_H_16_O_3_N_2_Na [M + Na]^+^ m/z, 331.10531; found, 3331.10440.

#### 2.2.9. 1-(3-Methoxyphenyl)-4-(4-methylbenzoyl)piperazine-2,5-dione (**9e**)

White solid, 86% yield, mp: 142–145 °C. ^1^H NMR (600 MHz, chloroform-d) δ: 7.59 (d, J = 8.2 Hz, 2H), 7.36 (t, J = 8.1 Hz, 1H), 7.24 (d, J = 8.0 Hz, 2H), 6.93–6.89 (m, 2H), 6.89–6.85 (m, 1H), 4.57 (s, 2H), 4.47 (s, 2H), 3.83 (s, 3H), 2.41 (s, 3H). ^13^C NMR (151 MHz, CDCl_3_) δ: 171.16, 166.91, 164.73, 160.40, 144.06, 140.46, 131.18, 130.28, 129.24, 129.19, 116.84, 113.21, 111.15, 55.60, 54.10, 48.55, 21.86. HRMS (ESI): calcd for C_19_H_19_O_4_N_2_ [M + H]^+^ m/z, 339.13393; found, 339.13260; C_19_H_18_O_4_N_2_Na [M + Na]^+^ m/z, 361.11588; found, 361.11490.

#### 2.2.10. 1-(4-Methoxyphenyl)-4-(4-methylbenzoyl)piperazine-2,5-dione (**9f**)

White solid, 87% yield, mp: 142–145 °C. ^1^H NMR (600 MHz, Chloroform-d) δ: 7.59 (d, J = 8.2 Hz, 2H), 7.31–7.18 (m, 4H), 7.01–6.92 (m, 2H), 4.56 (s, 2H), 4.44 (s, 2H), 3.83 (s, 3H), 2.41 (s, 3H). ^13^C NMR (151 MHz, CDCl_3_) δ: 171.22, 166.99, 164.85, 158.84, 144.03, 132.15, 131.23, 129.22, 129.19, 126.40, 114.83, 55.66, 54.45, 48.48, 21.86. HRMS (ESI): calcd for C_19_H_19_O_4_N_2_ [M + H]^+^ m/z, 339.13393; found, 339.13303; C_19_H_18_O_4_N_2_Na [M + Na]^+^ m/z, 361.11588; found, 361.11481.

#### 2.2.11. 1-(3,5-Dimethoxyphenyl)-4-benzoylpiperazine-2,5-dione (**9g**)

White solid, 85% yield, mp: 130–133 °C. ^1^H NMR (600 MHz, chloroform-d) δ: 7.70–7.65 (m, 2H), 7.61–7.53 (m, 1H), 7.45 (t, J = 7.8 Hz, 2H), 6.49 (d, J = 2.2 Hz, 2H), 6.43 (t, J = 2.2 Hz, 1H), 4.59 (s, 2H), 4.47 (s, 2H), 3.81 (s, 6H). ^13^C NMR (151 MHz, CDCl_3_) δ: 171.28, 166.82, 164.65, 161.41, 140.97, 134.16, 133.01, 128.94, 128.48, 103.56, 99.73, 55.70, 54.17, 48.47. HRMS (ESI): calcd for C_19_H_19_O_5_N_2_ [M + H]^+^ m/z, 355.12885; found, 355.12750; C_19_H_18_O_5_N_2_Na [M + Na]^+^ m/z, 377.11079; found, 377.10965.

#### 2.2.12. 1-(3,5-Dimethoxyphenyl)-4-(4-fluorobenzoyl)piperazine-2,5-dione (**9h**)

White solid, 88% yield, mp: 149–152 °C. ^1^H NMR (600 MHz, chloroform-d) δ: 7.74–7.68 (m, 2H), 7.12 (t, J = 8.5 Hz, 2H), 6.48 (d, J = 2.2 Hz, 2H), 6.43 (t, J = 2.2 Hz, 1H), 4.57 (s, 2H), 4.47 (s, 2H), 3.81 (s, 6H). ^13^C NMR (151 MHz, CDCl3) δ: 170.04, 166.89, 165.84 (166.47, 164.78, d, ^1^J = 254.9 Hz), 164.52, 161.39, 140.90, 131.76 (131.79, 131.73, d, ^3^J = 9.3 Hz), 130.19 (130.20, 130.18, d, ^4^J = 3.1 Hz, ), 115.74 (115.82, 115.67, d, ^2^J = 22.3 Hz), 103.53, 99.65, 55.69, 54.15, 48.55. ^19^F NMR (565 MHz, CDCl_3_) *δ*: -104.70. HRMS (ESI): calcd for C_19_H_18_O_5_N_2_F [M + H]^+^ m/z, 373.11943; found, 373.11835; C_19_H_17_O_5_N_2_FNa [M + Na]^+^ m/z, 395.10137; found, 395.10043.

#### 2.2.13. 1-(3,5-Dimethoxyphenyl)-4-(4-(trifluoromethyl)benzoyl)piperazine-2,5-dione (**9i**)

White solid, 78% yield, mp: 141–144 °C. ^1^H NMR (600 MHz, chloroform-d) δ: 7.74 (d, J = 8.2 Hz, 2H), 7.71 (d, J = 8.2 Hz, 2H), 6.48 (d, J = 2.2 Hz, 2H), 6.44 (t, J = 2.3 Hz, 1H), 4.63 (s, 2H), 4.48 (s, 2H), 3.81 (s, 6H). ^13^C NMR (151 MHz, CDCl3) δ: 170.02, 166.73, 164.22, 161.45, 140.82, 137.69, 134.05 (134.37, 134.15, 133.94, 133.72, q, ^2^J = 32.9 Hz), 128.95, 125.53 (125.56, 125.54, 125.51, 125.49, q, ^3^J = 3.8 Hz), 123.62 (126.32, 124.51, 122.71, 120.90, q, ^1^J = 273.0 Hz), 103.57, 99.74, 55.71, 54.14, 48.24. ^19^F NMR (565 MHz, CDCl_3_) *δ*: −63.11. HRMS (ESI): calcd for C_20_H_18_O_5_N_2_F_3_ [M + H]^+^ m/z, 423.11623; found, 423.11514; C_20_H_17_O_5_N_2_F_3_Na [M + Na]^+^ m/z, 445.0918; found, 445.09738.

#### 2.2.14. 1-(3,5-Dimethoxyphenyl)-4-(4-(tert-butyl)benzoyl)piperazine-2,5-dione (**9j**)

White solid, 70% yield, mp: 193–196 °C. ^1^H NMR (600 MHz, chloroform-d) δ: 7.64 (d, J = 8.4 Hz, 2H), 7.46 (d, J = 8.4 Hz, 2H), 6.49 (d, J = 2.0 Hz, 2H), 6.43 (t, J = 2.1 Hz, 1H), 4.57 (s, 2H), 4.47 (s, 2H), 3.81 (s, 6H), 1.34 (s, 9H). ^13^C NMR (151 MHz, CDCl_3_) δ: 171.16, 166.91, 164.73, 161.41, 156.99, 141.02, 131.04, 129.14, 125.51, 103.57, 99.75, 55.69, 54.20, 48.62, 35.29, 31.19. HRMS (ESI): calcd for C_23_H_27_O_5_N_2_ [M + H]^+^ m/z, 411.19145; found, 411.18982; C_23_H_26_O_5_N_2_Na [M + Na]^+^ m/z, 433.17339; found, 433.17227.

#### 2.2.15. 1-(3,5-Dimethoxyphenyl)-4-acetylpiperazine-2,5-dione (**9k**)

White solid, 63% yield, mp: 172–175 °C. ^1^H NMR (600 MHz, chloroform-d) δ: 6.42 (d, J = 2.2 Hz, 2H), 6.40 (t, J = 2.2 Hz, 1H), 4.56 (s, 2H), 4.46 (s, 2H), 3.78 (s, 6H), 2.62 (s, 3H). ^13^C NMR (151 MHz, CDCl_3_) δ: 171.19, 166.36, 164.27, 161.34, 140.93, 103.46, 99.70, 55.66, 54.74, 46.49, 27.11. HRMS (ESI): calcd for C_14_H_17_O_5_N_2_ [M + H]^+^ m/z, 293.11320; found, 293.11212; C_14_H_16_O_5_N_2_Na [M + Na]^+^ m/z, 315.09514; found, 315.09421.

#### 2.2.16. 1-(3,5-Dimethoxyphenyl)-4-(3-bromobenzoyl)piperazine-2,5-dione (**9l**)

White solid, 88% yield, mp: 192–195 °C. ^1^H NMR (600 MHz, chloroform-d) δ: 7.78 (s, 1H), 7.70–7.65 (m, 1H), 7.59–7.54 (m, 1H), 7.32 (t, J = 7.9 Hz, 1H), 6.48 (d, J = 2.2 Hz, 2H), 6.43 (t, J = 2.2 Hz, 1H), 4.58 (s, 2H), 4.47 (s, 2H), 3.80 (s, 6H). ^13^C NMR (151 MHz, CDCl_3_) δ: 169.78, 166.70, 164.31, 161.41, 140.86, 136.15, 135.71, 131.64, 129.96, 127.31, 122.46, 103.54, 99.73, 55.70, 54.08, 48.34. HRMS (ESI): calcd for C_19_H_18_O_5_ N_2_Br [M + H]^+^ m/z, 433.03936; found, 433.03812.

#### 2.2.17. 1-(3,5-Dimethoxyphenyl)-4-propionylpiperazine-2,5-dione (**9m**)

White solid, 60% yield, mp: 112–115 °C. ^1^H NMR (600 MHz, chloroform-d) δ: 6.42 (d, J = 2.0 Hz, 2H), 6.39 (t, J = 2.0 Hz, 1H), 4.57 (s, 2H), 4.45 (s, 2H), 3.78 (s, 6H), 3.00 (q, J = 7.2 Hz, 2H), 1.19 (t, J = 7.2 Hz, 3H). ^13^C NMR (151 MHz, CDCl_3_) δ: 175.24, 166.24, 164.41, 161.33, 140.94, 103.43, 99.66, 55.65, 54.85, 46.65, 32.81, 8.77. HRMS (ESI): calcd for C_15_H_19_O_5_N_2_ [M + H]^+^ m/z, 307.12885; found, 307.12772; C_15_H_18_O_5_N_2_Na [M + Na]^+^ m/z, 329.11079; found, 329.10980.

#### 2.2.18. 1-(3,5-Dimethoxyphenyl)-4-butyrylpiperazine-2,5-dione (**9n**)

White solid, 65% yield, mp: 136–139 °C. ^1^H NMR (600 MHz, chloroform-d) δ: 6.42 (d, J = 2.2 Hz, 2H), 6.40–6.38 (m, 1H), 4.56 (s, 2H), 4.45 (s, 2H), 3.78 (s, 6H), 2.95 (t, J = 7.4 Hz, 2H), 1.71 (q, J = 7.4 Hz, 2H), 0.99 (t, J = 7.4 Hz, 3H). ^13^C NMR (151 MHz, CDCl_3_) δ: 174.31, 166.25, 164.43, 161.33, 140.94, 103.43, 99.66, 55.65, 54.86, 46.57, 41.04, 17.95, 13.78. HRMS (ESI): calcd for C_16_H_21_O_5_N_2_ [M + H]^+^ m/z, 321.14450; found, 321.14340; C_16_H_20_O_5_N_2_Na [M + Na]^+^ m/z, 343.12644; found, 343.12558.

#### 2.2.19. 1-(3,5-Difluorobenzoyl)-4-(3,5-dimethoxyphenyl)piperazine-2,5-dione (**9o**)

White solid, 82% yield, mp: 182–185 °C. ^1^H NMR (600 MHz, chloroform-d) δ: 7.19–7.15 (m, 1H), 7.15–7.12 (m, 1H), 7.00 (tt, J = 8.6, 2.3 Hz, 1H), 6.47 (d, J = 2.2 Hz, 2H), 6.43 (t, J = 2.2 Hz, 1H), 4.58 (s, 2H), 4.48 (s, 2H), 3.81 (s, 6H). ^13^C NMR (151 MHz, CDCl3) δ: 168.94 (168.95, 168.93, 168.90, t, ^4^J = 3.3 Hz), 166.68, 164.07, 162.71 (163.58, 163.49, 161.91, 161.83, dd, ^1^J = 250.9, 12.1 Hz), 161.45, 140.78, 137.31 (137.37, 137.30, 137.24, t, ^3^J = 9.2 Hz), 111.83 (111.91, 111.87, 111.76, 111.72, dd, ^2^J = 21.4, 5.9 Hz), 108.10 (108.26, 108.09, 107.93, t, ^2^J = 25.2 Hz), 103.57, 99.81, 55.71, 54.06, 48.28. ^19^F NMR (565 MHz, CDCl_3_) *δ*: -108.13. HRMS (ESI): calcd for C_19_H_17_O_5_N_2_F_2_ [M + H]^+^ m/z, 391.11000; found, 391.10870; C_19_H_16_O_5_N_2_F_2_Na [M + Na]^+^ m/z, 413.09195; found, 413.09103.

#### 2.2.20. 1-(3,5-Dimethoxyphenyl)-4-pivaloylpiperazine-2,5-dione (**9p**)

White solid, 75% yield, mp: 180–183 °C. ^1^H NMR (600 MHz, chloroform-d) δ: 6.44 (d, J = 2.2 Hz, 2H), 6.41 (t, J = 2.2 Hz, 1H), 4.42 (s, 2H), 4.37 (s, 2H), 3.79 (s, 6H), 1.36 (s, 9H). ^13^C NMR (151 MHz, CDCl_3_) δ: 184.59, 166.61, 164.93, 161.38, 140.94, 103.48, 99.66, 55.68, 54.29, 49.42, 43.61, 27.32. HRMS (ESI): calcd for C_17_H_23_O_5_N_2_ [M + H]^+^ m/z, 335.16015; found, 335.15897; C_17_H_22_O_5_N_2_Na [M + Na]^+^ m/z, 357.14209; found, 357.14112.

#### 2.2.21. 1-(3,5-Dimethoxyphenyl)-4-(cyclopropanecarbonyl)piperazine-2,5-dione (**9q**)

White solid, 64% yield, mp: 155–158 °C. ^1^H NMR (600 MHz, chloroform-d) δ: 6.44 (d, J = 2.2 Hz, 2H), 6.40 (t, J = 2.2 Hz, 1H), 4.54 (s, 2H), 4.49 (s, 2H), 3.79 (s, 6H), 3.02–2.95 (m, 1H), 1.23–1.19 (m, 2H), 1.09–1.04 (m, 2H). ^13^C NMR (151 MHz, CDCl_3_) δ: 175.86, 166.96, 164.51, 161.34, 140.97, 103.45, 99.70, 55.67, 54.91, 47.14, 16.15, 11.92. HRMS (ESI): calcd for C_16_H_19_O_5_N_2_ [M + H]^+^ m/z, 319.12885; found, 319.16762; C_16_H_18_O_5_N_2_Na [M + Na]^+^ m/z, 341.11079; found, 341.10986.

#### 2.2.22. 1-(3,5-Dimethoxyphenyl)-4-(3-cyclopentylpropanoyl)piperazine-2,5-dione (**9r**)

White solid, 88% yield, mp: 141–144 °C. ^1^H NMR (600 MHz, chloroform-d) δ: 6.42 (d, J = 2.2 Hz, 2H), 6.39 (t, J = 2.2 Hz, 1H), 4.55 (s, 2H), 4.44 (s, 2H), 3.78 (s, 6H), 3.01–2.95 (m, 2H), 1.85–1.74 (m, 3H), 1.72–1.65 (m, 2H), 1.65–1.56 (m, 2H), 1.54–1.48 (m, 2H), 1.14–1.08 (m, 2H). ^13^C NMR (151 MHz, CDCl_3_) δ: 174.64, 166.21, 164.44, 161.31, 140.93, 103.42, 99.63, 55.64, 54.85, 46.62, 39.64, 38.47, 32.62, 30.71, 25.22. HRMS (ESI): calcd for C_20_H_27_O_5_N_2_ [M + H]^+^ m/z, 375.19145; found, 375.18997; C_20_H_26_O_5_N_2_Na [M + Na]^+^ m/z, 397.17339; found, 397.17230.

#### 2.2.23. 1-(3,5-Dimethoxyphenyl)-4-(2-methoxyacetyl)piperazine-2,5-dione (**9s**)

White solid, 40% yield, mp: 138–141 °C. ^1^H NMR (600 MHz, chloroform-d) δ: 6.41 (d, J = 2.2 Hz, 2H), 6.40 (t, J = 2.2 Hz, 1H), 4.62 (s, 2H), 4.60 (s, 2H), 4.46 (s, 2H), 3.78 (s, 6H), 3.49 (s, 3H). ^13^C NMR (151 MHz, CDCl_3_) δ: 171.78, 166.14, 163.81, 161.38, 140.86, 103.51, 99.76, 74.91, 59.62, 55.67, 54.48, 46.28. HRMS (ESI): calcd for C_15_H_19_O_6_N_2_ [M + H]^+^ m/z, 323.12376; found, 323.12238; C_15_H_18_O_6_N_2_Na [M + Na]^+^ m/z, 345.10571; found, 345.10474.

### 2.3. Biological Antioxidant Assays Used for the Study of Compounds **9a**–**9s**

#### 2.3.1. Cell Culture

SH-SY5Y cells were acquired from the National Cell Culture Collection. Cells were grown at 37 °C with 5% CO_2_ in full RPMI-1640 (Gibco, Waltham, MA, USA) media supplemented with 10% fetal bovine serum (FBS) (Biological Industries, Buenos Aires, Argentina).

#### 2.3.2. Measurement of Intracellular ROS Production

An active oxygen analysis kit (Beyotime Biotech, Nantong, China) was used to detect ROS generation according to the instructions. In brief, the treated cells were collected and washed with phosphate-buffered saline (PBS), with the addition of dichlorodihydrofluorescein diacetate (DCFH-DA,10 μM) serum-free Roswell Park Memorial Institute (RPMI)/1640 culture medium. Incubation took place at 37 °C for 30 min, and observation or flow cytometer analysis was performed using a fluorescence microscope.

#### 2.3.3. Determination of Mitochondrial Membrane Potential

To calculate the mitochondrial membrane potential, KGA604 (Key Gen Biotech) was used according to the manufacturer’s instructions. Cells were incubated with JC-1 working liquid for 20 min at 37 °C, washed three times with cold 1 × PBS, and then, the treated cells were observed or detected using flow cytometry under a fluorescence microscope.

#### 2.3.4. Cell Apoptosis

For apoptosis testing, an FITC Annexin V cell apoptosis-detection kit (556547, BD Biosciences, San Jose, CA, USA) was used. Briefly, the treated cells were collected and incubated in a binding solution with 5 μL Annexin V–fluorescein isothiocyanate (FITC) and 5 μL PI. Acea Biosciences (San Diego, CA, USA) measured the rate of apoptosis.

#### 2.3.5. Extraction of Cytoplasmic and Nuclear Proteins

A CW0199S nuclear slurry was used to extract the nuclear slurry separation kit (CW0199S, CWBIOTECH, Beijing, China). Cells were simply harvested, and 1 mL NC Buffer A was added after centrifugation, mixed for 5 s, and incubated for 20 min on ice. Then, 55 μL NC buffer B was stirred in, and the solution was incubated for 1 min on ice. After being personalized for 15 min (4 °C, 12,000 rpm), the liquid was collected and stored. Then, 500 μL of NC Buffer C was added to the precipitate obtained in the preceding procedures, mixed for 5 s, and incubated on ice for 40 min, rotating for 15–30 s every 10 min. After being personalized for 15 min (4 °C, 12,000 rpm), the liquid was collected and preserved.

#### 2.3.6. Immunofluorescence Staining

In brief, the cells were incubated in 4% additional formaldehyde for 15 min before being washed three times with PBS and permeabilized with 0.1% Triton X-100. After three washes with PBS, the membrane was blocked with 4% bovine serum albumin for 30 min before being incubated overnight with the Nrf2 antibody at 4 °C. The next day, PBS was rinsed three times, and Alexa Fisher Science (Thermo Fisher Science, Waltham, MA, USA) was used. The duration of incubation was 5 min. To obtain pictures, a microscope (DMI8, Leica, Wetzlar, Germany) was used.

#### 2.3.7. SiRNA Transfection

China Guangzhou Nuclear Biological Co., Ltd. (Guangzhou, China) designed and synthesized small interfering RNA (siRNA) oligonucleotides for IL-6, Nrf2 and negative-control siRNA (NC-si). The specific operation method can be found in the literature [20]. In short, according to the instructions, the siRNA mixture was introduced into SH-SY5Y cells. After 48 h, the cells were treated with H_2_O_2_ and 9r, and Western markers were used to detect changes in related proteins. The following siRNA sequences were used: IL-6-siRNA: 5′-GCTACGTGATGAGAGGGA-3′; Nrf2-siRNA: 5′-GCATCTCAGCCCCCTGAA-3′.

#### 2.3.8. Western Blotting

In brief, proteins were collected using lysis buffer (Cat No. P0013B, Beyotime, Shanghai, China), and the protein concentration was measured using a BCA Protein Assay Kit (PC0020, Solarbio Science & Technology, Beijing, China). Before incubation with the antibody, equal quantities of protein samples were separated by denaturing sodium dodecyl sulfate-polyacrylamide gel electrophoresis (SDS-PAGE) and transferred to polyvinylidene fluoride (PVDF) membranes. A ChemiDoc MP Imaging System was used to image the blots (Bio-Rad, Hercules, CA, USA). p-SAPK/JNK (Thr183/Tyr185; #4685), SAPK/JNK (#9252), p-Src (Ser17; #12432), p-Stat3 (Tyr705; #9145), mTOR (2065AP), p-mTOR (Ser2448; #5536), p-Akt (Ser473; #4060), p-GSK3β (Ser9; #5558), PARP (#9532) and cleaved-PARP (#94885) were provided by Cell Signaling Technology. Src (11097-1-AP), AKT (11097-1-AP), Stat3 (10253-2-AP), ERK1/2 (16443-1-AP), GSK3β (#12456), GAPDH (10494-1-AP), P53 (10442-1-AP), Bcl2 (10068-1-AP), Bax (50599-2-19), IL-6 (21865-1-AP), Nrf2 (16396-1-AP) and HO-1 (10701-1-AP) were purchased from Proteintech. Immobilon^®^ PVDF membranes were provided by Thermo Fisher Scientific (Waltham, MA, USA) and Merck KGaA (Darmstadt, Germany), respectively.

#### 2.3.9. Statistical Analysis

All experiments were carried out in triplicate and repeated three times. The data are reported as the mean ± standard deviation (SD). Statistical analysis was performed by applying Kruskal–Wallis univariate analysis of variance (ANOVA) in SPSS. The treatment effect was considered significant at *p* < 0.05.

## 3. Results and Discussion

### 3.1. Chemistry

In our present study, precursors **7a**–**e** [19] were synthesized from aromatic amines **1a**–**e**. Compounds **1a**–**e** were allowed to react with ethyl bromoacetate via a nucleophilic substitution reaction to give intermediates **2a**–**e**. The treatment of **2a**–**e** with chloroacetyl chloride under phase-transfer catalysis conditions gave amides **3a**–**e**. Compounds **3a**–**e** were then treated with excess sodium azide in dimethylformamide (DMF) solution at 90 °C to produce azide compounds **4a**–**e** in excellent yields. Subsequently, the intramolecular aza-Wittig reaction was conducted with substrates **4a**–**e**. Intermediates **4a**–**e** were treated with triphenylphosphine in tetrahydrofuran (THF) at room temperature to give compounds **5a**–**e**. Intermediates **6a**–**e** were obtained from compounds **5a**–**e** via the spontaneous cyclization reaction. Next, intermediates **6a**–**e** were stirred in moist THF (not anhydrous), resulting in enol ether cleavage and rearrangement to give **7a**–**e**. Finally, compounds **7a**–**e** reacted with the corresponding chloride (**8**) to form target compounds **9a**-**s** (Figure 1). The structures of all the synthetic compounds are summarized in Table 1.

### 3.2. Biological Evaluation

#### 3.2.1. Cell Viability Assay

Before the examination of the antioxidative activities, the SH-SY5Y cells were treated with the tested compounds at different concentrations to determine their cytotoxicity. As shown in Appendix A, almost all the tested compounds at 80 μM showed no obvious cytotoxicity to SH-SY5Y cells. In contrast, most compounds at 20 μM could significantly promote cell proliferation. Therefore, 20 μM of the tested compounds were selected for further evaluation of their potential antioxidant activities. As shown in Table 2, most of the derivatives showed better cytoprotective activity when compared with the lead compound Syringaresinol. In addition, some derivatives (**9d**, **9e**, **9o** and **9r**) exhibited similar cytoprotective activity or better activity than that of the positive control Neohesperidin. Compound **9r**, with the best activity, significantly increased the viability of H_2_O_2_-treated SH-SY5Y cells from 32.85% (H_2_O_2_-treated group) to 70.19% (**9r**-treated group). Therefore, we chose to further explore the possible antioxidative mechanism of compound **9r**.

#### 3.2.2. Compound **9r** Exhibited Cytoprotective Effects by Partially Restoring Cell Viability and Ameliorating Morphology

The viability of SH-SY5Y cells was further evaluated after treatment with various concentrations of **9r** to investigate its dose-dependent antioxidative activity. As shown in Figure 2A, Compound **9r** (5–20 μM) dose-dependently protected against H_2_O_2_-induced oxidative stress and promoted the viability of H_2_O_2_-treated SH-SY5Y cells. Compound **9r** showed better antioxidative activity than the lead compound syringaresinol (Figure 2B). In addition, similar cytoprotective activities against H_2_O_2_-induced oxidative stress were observed in **9r**- and neohesperidin (positive control)-pretreated SH-SY5Y cells (Figure 2B). The results of the morphological alterations showed that H_2_O_2_ treatment dramatically decreased the number of living cells and resulted in cell contraction and nuclear fragmentation (Figure 2C). However, **9r** significantly prevented the cell morphological changes caused by H_2_O_2_ in a dose-dependent manner. These results suggested that **9r** possessed a cytoprotective effect against oxidative damage caused by H_2_O_2_ treatment in SH-SY5Y cells.

#### 3.2.3. **9r** Attenuated H_2_O_2_-Induced Cell Apoptosis in SH-SY5Y Cells

H_2_O_2_-induced oxidative stress eventually results in a dramatic increase in cell apoptosis [21]. To investigate whether the cytoprotective effect of **9r** was associated with the mediation of cell apoptosis, flow cytometry was used to detect the effect of **9r** on H_2_O_2_-induced apoptosis using an Annexin V–fluorescein isothiocyanate (FITC) apoptosis-detection kit. Figure 3 demonstrates that the apoptosis rate in the H_2_O_2_-treated group (29.5%) was substantially greater than the apoptosis rate in the control group (1.28%); however, **9r** pretreatment for 2 h dose-dependently inhibited H_2_O_2_-induced apoptosis. To further determine the inhibitory activity of **9r** on cell apoptosis, we next examined the influence of **9r** on proteins involved in the apoptotic pathways using Western blotting. As indicated in Figure 4, exposure to H_2_O_2_ dramatically increased the related proapoptotic proteins (p53, Bax, Cl-PARP) while decreasing the antiapoptotic protein Bcl2. However, **9r** pretreatment abolished the H_2_O_2_-induced alterations in the related apoptotic proteins, which was consistent with our prior findings. The data suggested that the cytoprotective activity of **9r** was partially attributed to the inhibitory effects of the apoptotic pathway.

#### 3.2.4. Compound **9r** Attenuated Mitochondrial Dysfunction Caused by H_2_O_2_ Treatment

As previously mentioned, mitochondrial dysfunction is one of the major causes of H_2_O_2_-induced cell apoptosis [22]. Therefore, the effect of **9r** on mitochondrial function in H_2_O_2_-treated SH-SY5Y cells was examined via flow cytometry and inverted fluorescence microscopy using JC-1 dye. As shown in Figure 5, the mitochondrial membrane potential was significantly elevated in the H_2_O_2_-treated group compared with the control group. However, **9r** pretreatment counteracted the effects of H_2_O_2_ on mitochondrial membrane potential (Figure 5). The results suggested that **9r** could resist the mitochondrial dysfunction caused by H_2_O_2_.

#### 3.2.5. **9r** Decreased the Generation of ROS in H_2_O_2_-Treated SH-SY5Y Cells

Mitochondrial dysfunction often dramatically increases the production of ROS and subsequently promotes cell apoptosis through the mitochondrial apoptosis pathway [23]. Given that **9r** can protect the mitochondrial membrane potential, we speculate that **9r** can also resist oxidative stress by reducing the production of ROS. Therefore, the effect of **9r** on intracellular ROS production after H_2_O_2_ treatment was examined. As shown in Figure 6, H_2_O_2_ treatment significantly increased ROS production compared to the control group. Nevertheless, these effects were partly neutralized by **9r** pretreatment for 2 h. Accordingly, immunofluorescence analysis also indicated that the increased green fluorescence caused by H_2_O_2_ was steadily downregulated by **9r** pretreatment, suggesting that **9r** reduced H_2_O_2_-induced ROS production in a dose-dependent manner. In summary, **9r** may counteract the oxidative injury caused by H_2_O_2_ by reducing ROS generation.

#### 3.2.6. The Antioxidant Capacity of **9r** Is Mediated by the Nrf2/HO-1 Signaling Pathway

In neurodegenerative diseases such as Parkinson’s disease, Alzheimer’s disease and cerebral ischemia, activated Nrf2 is able to ameliorate various pathological processes, such as mitochondrial dysfunction and oxidative stress, thus showing antioxidant benefits [24,25]. Nrf2 has been reported to initiate the transcription of relevant protective genes against oxidative stress after translocation from the cytoplasm to the nucleus by binding to antioxidant response elements. These elements include heme oxygenase-1 (HO-1), an important antioxidant enzyme that is one of the key downstream targets for Nrf2 to exert cytoprotective effects [26]. Therefore, after confirming the protective effect of **9r** on mitochondria, we examined its effect on the nuclear translocation of Nrf2. As depicted in Figure 7A–D, Western blot analysis indicated that **9r** significantly prevented the reduction in Nrf2 nucleation caused by H_2_O_2_. This effect of **9r** on Nrf2 was also confirmed by immunofluorescence analysis (Figure 7E). These results demonstrated that the Nrf2/HO-1 signaling pathway plays a very important role in the antioxidant activity of **9r**.

#### 3.2.7. **9r** Exerted Antioxidant Activity Depending on the Activation of Nrf2

Because Nrf2 was confirmed to be associated with the antioxidative activity of **9r**, we further determined whether Nrf2 was essential to the inhibitory effect of **9r** on oxidative stress. As shown in Figure 8, Nrf2 was knocked down by siRNA transfection. The protein levels of Nrf2 and its downstream proteins, including HO-1 and IL-6, were examined via Western blotting. Accordingly, Nrf2 knockdown reduced the expression of HO-1 and IL-6 (Figure 8). In contrast, these proteins were maintained at relatively high levels after **9r** pretreatment, suggesting that the upregulation of Nrf2 would lead to a corresponding increase in IL-6 (Figure 8). More importantly, **9r** pretreatment decreased markers of cell apoptosis, including cleaved poly-ADP ribose polymerase (PARP) and p53, but these effects were compromised by Nrf2 knockdown (Figure 8). The results indicated that **9r** might resist H_2_O_2_-induced oxidative stress by targeting the Nrf2/IL-6 positive-feedback loop.

#### 3.2.8. **9****r** Promotes Cell Survival by Regulating the IL-6/Src/Stat3 Signaling Pathway

In previous cell viability tests, **9r** at 20 μM was found to show significant pro-proliferative activity. Therefore, we believe that the antiapoptotic properties of **9r** may also be related to the promotion of cell survival, beyond its antioxidant activity. Previous studies have shown that the activation of IL-6 and its downstream signaling pathways could promote cell survival [27]. To verify this hypothesis, we examined the effect of **9r** on IL-6 downstream components. In Figure 9, the protein levels of IL-6, p-Stat3 and p-Src were significantly reduced after H_2_O_2_ treatment. However, the related proteins in H_2_O_2_-treated SH-SY5Y cells were elevated by **9r** pretreatment. These data suggested that **9r** can promote cell growth by regulating the IL-6/Src/Stat3 signaling pathway.

#### 3.2.9. **9r** Regulates the PI3K and MAPK Signaling Pathways to Promote Cell Survival

IL-6 has also been reported to stimulate the Src-PI3K signaling cascade, which then activates downstream signaling pathways such as PI3K/Akt and mitogen-activated protein kinase (MAPK) [28,29,30,31,32], thereby promoting cell survival. We therefore examined whether the above signaling pathways were involved in the role of **9r** in promoting cell growth.

The effect of the PI3K signaling pathway on **9r**-promoted cell growth was first explored. Compared to the control group, the phosphorylation levels of PI3K, AKT, mTOR and GSK3β were significantly reduced in the H_2_O_2_-treated group. However, **9r** pretreatment significantly restored the phosphorylation levels of the above proteins (Figure 10). These data implied that PI3K signaling contributed to the prosurvival properties of **9r**.

In terms of the MAPK pathway, higher phosphorylation levels of JNK and P38, as well as lower phosphorylation levels of ERK, were observed in the H_2_O_2_-treated group than in the control group. Instead, **9r** reversed the changes in the above proteins induced by H_2_O_2_ exposure (Figure 11). These results suggested that **9r** could also regulate the MAPK signaling pathway to promote cell survival. Taken together, we concluded that **9r** exhibited cytoprotective properties against H_2_O_2_-induced oxidative stress injury to SH-SY5Y cells via the Nrf2 antioxidant pathway and IL-6 prosurvival signaling.

#### 3.2.10. **9r** Regulates Oxidative Stress via an IL-6/Nrf2 Positive-Feedback Loop

To further verify the role of the Nrf2/IL-6 positive-feedback loop in the antioxidative activity of **9r**, siRNA was employed to knock down IL-6 to confirm the bidirectional regulation between IL-6 and Nrf2 and to explore the effect of **9r** on IL-6 and its downstream signaling pathways. Indeed, the results revealed that Nrf2 expression was correspondingly reduced by IL-6 knockdown, suggesting bidirectional regulation between IL-6 and Nrf2. Moreover, the downregulation of IL-6 decreased the levels of the Nrf2 downstream enzymes HO-1 and the IL-6 downstream target molecules p-STAT3, p-Src, p-ERK and p-PI3K (Figure 12). Importantly, the effects of IL-6 knockdown on Nrf2- and IL-6-related downstream pathways were offset, to some extent, by 20 μM **9r** pretreatment (Figure 12). Based on the above results, we concluded that **9r** significantly suppressed H_2_O_2_-induced oxidative stress depending on the Nrf2/IL-6 positive-feedback loop.

## 4. Conclusions

In the present study, a series of 1,4-disubstituted piperazine-2,5-diones were designed and synthesized, and their biological activities were evaluated. Among them, compound **9r** exhibited the best antioxidative activity. Further mechanistic studies showed that **9r** inhibited apoptosis and promoted cell survival via reducing ROS production and by stabilizing the mitochondrial membrane potential. The cytoprotective properties of **9r** against H_2_O_2_-induced oxidative stress injury in SH-SY5Y cells are dependent on the Nrf2/IL-6 positive-feedback loop (Figure 13). These results implied that compound **9r** has the potential to be a novel antioxidant for the treatment of neurodegenerative diseases caused by oxidative stress.

## Data Availability

All data are included within the article and Appendix A.

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
