# Peer review of "Design, Synthesis and Evaluation of Novel 1,4-Disubstituted Piperazine-2,5-dione Derivatives as Antioxidants against H2O2-Induced Oxidative Injury via the IL-6/Nrf2 Loop Pathway"

_antioxidants, 2022, doi:10.3390/antiox11102014_

Round 1
Reviewer 1 Report
In this manuscript by Fan and co-workers have synthesized several new heterocyclic compounds and studied their ability to prevent oxidative stress in SH-SY5Y cells. The authors have also investigated and identified the mechanism of action as being stabilization of the mitochondrial membrane. I do not see any concerns regarding the biological studies. Regarding the synthetic portion of the work:
1. The authors need to proofread their experimental especially – nothing that should be subscripted is subscripted.
2. Some of the compounds which have been prepared are known compounds. The authors should cite manuscripts that have previously shown these molecules as well as ensure the NMR data is consistent.
3. Regarding the NMR data, it is hard to gauge the transcriptions because as far as I can tell the actual spectra have not been provided. However, I can tell that the authors need to revise their carbon NMR transcriptions. For NMR active heteroatoms (such as fluorine) it is appropriate to provide the coupling constants for carbon signals that are split in the same way it is done for the proton NMR. Some peaks (especially for CF3 groups) may be partially overlapping or of low intensity, but the authors should be able to reason from center of mass where the peak is and what the multiplicities are.
4. Heteroatom NMR (19F) is not provided.
Reviewer 2 Report
Comments
The manuscript "Design, synthesis and evaluation of novel 1,4-disubstituted piperazine-2,5-dione derivatives as antioxidants against H2O2-induced oxidative injury via the IL-6/Nrf2 loop pathway" revealed that some of the newly synthesized compounds exhibited antioxidative activities, and compound 9r showed better antioxidative activity than the lead compound syringaresinol. Experiments were performed by authors to show compound 9r decreased ROS production and stabilized mitochondrial membrane potential to restrain cell apoptosis and promoted cell survival via an IL-6/Nrf2 positive feedback loop.
Authors used the scaffold hopping strategy to design novel 1,4-disubstituted piperazine-2,5-dione derivatives from natural bisepoxylignans scaffold. After simplifying the scaffold, a series of 1,4-disubstituted piperazine-2,5-dione derivatives could be easily synthesized in large quantities.
1. However, if more explanation for using the 2,5-piperazinedione but not other heterocycle replacement, will be better for this high quality of journal.
2. From the result shown in Fig. 2A, it is difficult to present that compound 9r dose-dependently protected against H2O2-induced oxidative stress and promoted the viability of H2O2-treated SH-SY5Y cells.
